# Ensemble Object Detection Methodology for Automated Detection of Inflammatory Cells in Kidney Biopsies

**Gunjan Deotale**                                     GUNJAN.DEOTALE@AIRAMATRIX.COM
**Abhishek Ambast**                                    ABHISHEK.AMBAST@AIRAMATRIX.COM
**Lavish Ramchandani**                         LAVISH.RAMCHANDANI@AIRAMATRIX.COM
**Dev Kumar Das**                                      DEVKUMAR.DAS@AIRAMATRIX.COM
**Tijo Thomas**                                            TIJO.THOMAS@AIRAMATRIX.COM
*AIRA MATRIX Private Limited, Mumbai, India*

## Abstract

Automated detection of inflammatory cells in kidney biopsies is essential for kidney disease diagnosis. To address this, we participated in the Machine-learning for Optimal detection of iNflammatory cells in KidnEY (MONKEY) challenge, where the main challenges were to detect inflammatory cells and further classify them as monocyte and lymphocyte. We employed an ensemble of DETR and YOLOv5-L object detection models, achieving the 3rd place on both leaderboards with Free Response Receiver Operating Characteristic (FROC) scores of 0.3517 (Task 1) and 0.4471/0.1906 (Task 2). Our approach demonstrated the power of combining transformer-based and convolutional architectures to enhance diagnostic precision in digital pathology, offering a cost-effective alternative to immunohistochemistry (IHC) staining while advancing transplant rejection analysis.

**Keywords:** Kidney Biopsy, Inflammatory Cell, Object Detection, Digital Pathology

## 1. Introduction

Inflammation in kidney transplant biopsies , particularly the presence of mononuclear leukocytes such as monocytes and lymphocytes, is a critical indicator of rejection. Manual assessment of these cells in whole slide images (WSIs) is time consuming, prone to inter-observer variability, and limited in its ability to capture subtle spatial patterns of infiltration (Litjens et al., 2017) . Automated detection and classification of inflammatory cells could revolutionize this process by improving diagnostic consistency, reducing pathologist workload, and uncovering quantitative insights into transplant outcomes. One solution is to use immunohistochemistry (IHC) stained images, where the subtle differences between lymphocytes and monocytes are easily detectable to the human eye. Hermsen et al. (2022) developed a CNN model to detect lymphocytes in immunohistochemistry (IHC) stained slides, specifically targeting CD3 positive cells. However, IHC staining comes with its associated costs. To bridge this gap, we participated in the Monkey Challenge (Studer et al., 2025) , focusing on the automated detection and classification of inflammatory cells in PAS-stained kidney biopsies. The challenge provided us with associated IHC-stained images and co-registered PAS-stained WSIs. Leveraging the inherent details coming from the IHC data, and training a model to detect corresponding inflammatory cells, could provide a huge advantage to overcome cost barriers along with reducing manual effort in annotation. The challenge consisted of two tasks: Detection of mononuclear, inflammatory cells (mononuclear leukocytes (MNLs)) and further classification of inflammatory cells as monocytes and lymphocytes.

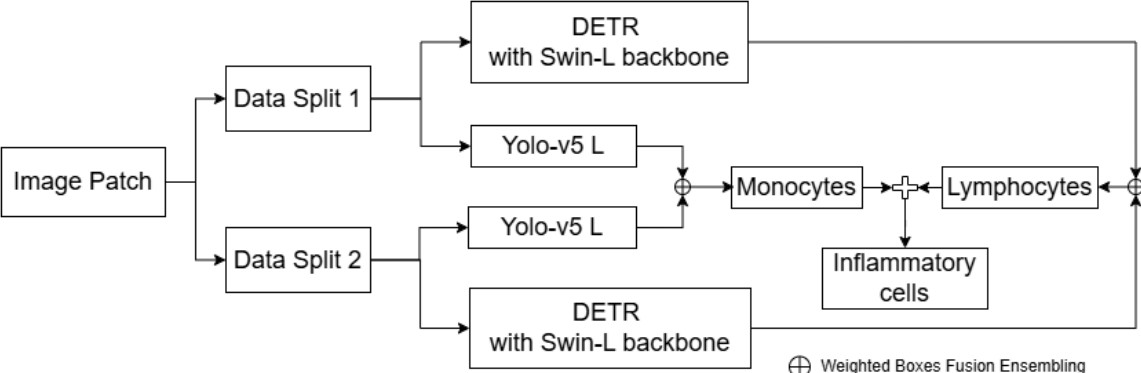

Figure 1: Workflow of the inflammatory cell detection methodology.

## 2. Materials and Methods

### 2.1. Dataset

The challenge dataset was curated from 153 WSIs collected across six different pathology departments, reflecting diverse staining and scanning protocols. The organizers annotated a total of 231 regions of interest (ROIs) with dot annotations for monocytes and lymphocytes. The training set consisted of 81 WSIs. We extracted 7566 tiles from PAS stained images of size $512 \times 512$ at 40X magnification for training.

### 2.2. Model training and ensembling

In this study, we employed two state-of-the-art object detection models: DEtection TRansformer (DETR) (Carion et al., 2020) with Swin-L Backbone (Liu et al., 2021) and YOLOv5-L (Khanam and Hussain, 2024), to evaluate their performance in detecting target objects. To fully leverage the dataset for training, we created two distinct splits and trained both models separately, then assembled their outputs. The DETR model utilized a cosine annealing learning rate scheduler with an initial learning rate of 0.0008, whereas YOLOv5-L adopted a cosine learning rate with plateau, starting at 0.003 with a reduction factor of 0.2. For data augmentation, both models were subject to geometric transformations, color and contrast adjustments, and mixing techniques to enhance model generalization. Both models employed the AdamW stochastic optimization method and utilized Binary Cross Entropy (BCE) loss for object detection loss calculation, ensuring robust handling of object presence probability. Quality focal loss was used as classification loss for DETR, while the cross-entropy loss was used for yolov5-L. Bounding box regression was handled using Smooth L1 loss in DETR , whereas YOLOv5-L leveraged Generalized Intersection over Union (GIoU) loss. To ensemble the models, we used the Weighted boxes fusion (WBF) method (Solovyev et al., 2021). Unlike NMS and soft-NMS methods that simply remove bounding boxes with intersection-over-union (IoU) higher than a threshold value, the proposed WBF method uses confidence scores of all proposed bounding boxes to construct fused boxes. This method significantly improved the quality of the output bounding boxes of the assembled models. The experiments were conducted on a high-performance system

Table 1: Performance comparison of different architectures.

| Architecture | rtmDet | DiffusionDet | Centernet | DDQ | DETR | yolov5-L | Ensemble |
|---|---|---|---|---|---|---|---|
| mAP * | 0.54/0.51 | 0.52/0.48 | 0.48/0.47 | 0.52/0.47 | 0.61/0.6 | 0.6/0.63 | 0.61/0.63 |
| FROC Score ** | 0.33/0.34/0.08 | 0.31/0.33/0.075 | 0.29/0.32/0.06 | 0.305/0.33/0.06 | 0.36/0.39/0.09 | 0.33/0.36/0.14 | 0.36/0.38/0.13 |

with Ubuntu 20.04.6 LTS, CUDA version 11.7, NVIDIA RTX A6000 GPU (48 GB VRAM), 55 GB CPU RAM, Python version 3.9.17, and PyTorch version 2.0.1.

## 3. Results and Discussion

We experimented with different architectures during the validation phase to identify the most effective models for inflammatory cell detection. These included rtmDet (Lyu et al., 2022), DETR (Carion et al., 2020), YOLOv5-l (Khanam and Hussain, 2024), Centernet (Duan et al., 2019), DDQ (Zhang et al., 2023) and diffusion det (Chen et al., 2023) . The models were tested on the validation set created using the intial 81 WSIs in the training set as well as the validation set utilized during the live leaderboard phase. As seen in Table 1, DETR performed the best for Lymphocyte detection with 0.61 Mean Average Precision (mAP) , while Yolov5-l gave the highest mAP of 0.63 for Monocyte detection on the validation dataset.

Our approach (Figure 1) leveraged DETR for lymphocyte detection, capitalizing on its hierarchical vision transformer architecture to handle dense lymphocyte clusters, and YOLOv5-l for monocyte detection, utilizing its efficient convolutional framework for sparse objects. As shown in Table 2, we ranked 3rd on both leaderboards in the competition with FROC score of 0.3517 for inflammatory cell detection in Task 1, and FROC scores of 0.4471 and 0.1906 for detection of lymphocyte and monocyte respectively in Task 2.

## 4. Conclusion

The ensemble of DETR and YOLOv5 improved the detection of all inflammatory cells, as validated by the FROC scores in Table 1. This dual-methodology strategy demonstrated the complementary strengths of transformer-based and convolutional architectures. The challenge established a robust benchmark for automated inflammatory cell detection in kidney biopsies. While top solutions, including ours, demonstrated promise, clinical deployment requires further refinement to enhance sensitivity and specificity beyond current thresholds.

Table 2: FROC scores of challenge winners for both tasks.

| Rank | Task 1 (Inflammatory cell) | Task 2 (Lymphocyte/Monocyte) |
|---|---|---|
| 1 | TIAKong [0.3930] | InstanSeg [0.4515/0.2626] |
| 2 | InstanSeg [0.3875] | TIAKong [0.4624/0.2392] |
| 3 | **Ours (Aira Matrix) [0.3517]** | **Ours (Aira Matrix) [0.4471/0.1906]** |

---

*The mAP was evaluated on a validation set derived from the 81 training WSIs, reported in the Lymphocyte/Monocyte format.*

**The FROC Score was evaluated on the challenge validation set, reported in Inflammatory cell/Lymphocyte/Monocyte format.*

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
