# OpenReview forum: "Ensemble Object Detection Methodology for Automated Detection of Inflammatory Cells in Kidney Biopsies"
_MIDL.io/2025/Short_Papers — MIDL 2025 - Short Papers_

### Official Review · Reviewer_KFu6 · 2025-04-25

**Rating:** 4
**Confidence:** 5

**Summary:**

This study presents an ensemble approach combining DETR and YOLOv5-L models for automated detection and classification of inflammatory cells in kidney biopsy WSIs. By training on PAS-stained images and leveraging IHC annotations, the method achieved strong results in the MONKEY challenge, securing 3rd place in both detection and classification tasks.

**Strengths:**

A major strength of the approach is the complementary use of transformer-based (DETR) and convolutional (YOLOv5-L) architectures, maximizing detection performance across both dense and sparse cell distributions. The use of Weighted Boxes Fusion further refined predictions beyond traditional NMS techniques.

**Weaknesses:**

Despite promising leaderboard results, the method's sensitivity and specificity still fall short of clinical deployment standards, and the approach may struggle to generalize across broader staining and imaging variations without additional domain adaptation strategies.

---

### Decision · Program_Chairs · 2025-05-01

Accept